



# On the influence of underlying elevation data on Sentinel-5 Precursor satellite methane retrievals over Greenland

Jonas Hachmeister[1], Oliver Schneising[1], Michael Buchwitz[1], Alba Lorente[2], Tobias Borsdorff[2], John P. Burrows[1], Justus Notholt[1], and Matthias Buschmann[1]

[1]Institute of Environmental Physics (IUP), University of Bremen FB1, Bremen, Germany
[2]Netherlands Institute for Space Research, SRON, Leiden, the Netherlands

**Correspondence:** J. Hachmeister (jonas_h@iup.physik.uni-bremen.de)

**Abstract.** The Sentinel-5 Precursor (S5P) mission was launched on October 2017 and has since provided data with high spatio-temporal resolution using its remote sensing instrument, the TROPOspheric Monitoring Instrument (TROPOMI). The latter is a nadir viewing passive grating imaging spectrometer. The mathematical inversion of the TROPOMI data yields retrievals of different trace gas and aerosol data products. The column-averaged dry air mole fraction of methane ($XCH_4$) is the product
of interest to this study. The daily global coverage of the atmospheric methane mole fraction data enables the analysis of the methane distribution and variation on large scales and also to estimate surface emissions. The spatio-temporal high-resolution satellite data are potentially particularly valuable in remote regions, such as the Arctic, where few ground stations and in-situ measurements are available. In addition to the operational Copernicus S5P total-column averaged dry air mole fraction methane data product developed by SRON, the scientific TROPOMI/WFMD algorithm data product v1.5 (WFMD product)
was generated at the Institute of Environmental Physics at the University of Bremen. In this study we focus on the assessment of both S5P $XCH_4$ data products over Greenland and find that spatial maps of both products show distinct features along the coast lines. Anomalies up to and exceeding $100\,\mathrm{ppb}$ are observed and stand out in comparison to the otherwise smooth changes in the methane distribution. These features are more pronounced for the operational product compared to the WFMD product. The spatial patterns correlate with the difference of the GMTED2010 digital elevation model (DEM) used in the retrievals to
a more recent topography data set indicating that inaccuracies in the assumed surface elevation are the origin of the observed features. These correlations are stronger for the WFMD product. In order to evaluate the impact of the topography dataset on the retrieval we reprocess the WFMD product with updated elevation data. We find a significant reduction of the localized features when GMTED2010 is replaced by recent topography data over Greenland based on ICESat-2 data. This study shows the importance of the chosen topography data on retrieved dry air mole fractions. Use of a precise and up-to-date as possible
DEM is advised for all S5P data products as well as for future missions which rely on DEM as input data. A modification based on this study is planned to be introduced to the next version of the WFMD data product.



# 1 Introduction

Following the launch of the Sentinel-5 Precursor (S5P) satellite mission, carrying the TROPOspheric Monitoring Instrument (TROPOMI), atmospheric measurements with unprecedented spatio-temporal resolution have become available. TROPOMI is a nadir viewing passive grating imaging spectrometer with a push broom configuration. Its near polar, sun-synchronous orbit provides a daily coverage of the Earth. While for some gases vertical columns are retrieved, the algorithms for methane ($CH_4$) retrieve the column-averaged dry air mole fractions (denoted $XCH_4$).

Methane is a globally well-distributed greenhouse gas and one of the most important drivers of climate change with a radiative forcing of $0.61\ \mathrm{Wm^{-2}}$ (Etminan et al., 2016) and an atmospheric lifetime of around 9 years (Masson-Delmotte et al., 2021). The concentration of $CH_4$ has increased by 156% between 1750 and 2019 reaching $1866\pm3.3\ \mathrm{ppb}$ in 2019 (Masson-Delmotte et al., 2021). Large amounts of soil organic carbon (SOC) are stored in the Arctic permafrost regions (ca. 1300 Pg) of which roughly $800\ \mathrm{Pg}$ is perennially frozen (Hugelius et al., 2014). Further warming of the Arctic may lead to increased permafrost degradation and rapid SOC loss (Plaza et al., 2019) by the release of carbon dioxide ($CO_2$) and/or methane. Monitoring emissions from the Arctic permafrost region is hence important but challenging as a result of the low surface reflectance of the ocean, ice and snow surface and the spatial extent. While ground-based and aircraft campaigns deliver vital information on a local or regional scale, the retrieved data products from satellite sensors yield potentially greater spatio temporal coverage.

At the moment three S5P/TROPOMI methane retrieval algorithms exist: The operational Copernicus S5P $XCH_4$ algorithm developed by SRON (Hu et al., 2016), the scientific Weighting Function Modified Differential Optical Absorption Spectroscopy (WFMD) algorithm developed at the Institute of Environmental Physics at the University of Bremen (Schneising et al., 2019) and the scientific algorithm of SRON (Lorente et al., 2021). In this work, we investigate the operational product (V01, Apituley et al., 2017) and the TROPOMI/WFMD product (v1.5, Schneising, 2021). Both yield the geolocated $XCH_4$ and auxiliary information.

In Sect. 2 we introduce the datasets used in this work, Sect. 3 describes the methods used in our analysis and Sect. 4 contains the results of our research. We finish with Sect. 5 where we present our conclusions.

# 2 Datasets

## 2.1 Sentinel-5 Precursor Mission

The Sentinel-5 Precursor satellite was launched on 13 October 2017 and has since delivered high quality data from its only scientific instrument, TROPOMI, which is a nadir viewing passive grating imaging spectrometer. Combined with a near-polar, sun-synchronous orbit, the swath width of 2600 km provides daily coverage of the Earth. Due to the orbit geometry and swath overlap multiple observations per day are possible in the polar regions. The instrument comprises four spectrometers measuring radiances in the ultraviolet (267–332 nm), ultraviolet-visible (305–499 nm), near infrared (661–786 nm) and short-wave infrared (2300–2389 nm) spectral range (Veefkind et al., 2012). The spatial resolution depends on the bands and is $7\times3.5$ $\mathrm{km^2}$ for the near infrared (NIR) bands (updated to $5.5\times3.5\ \mathrm{km^2}$ in August 2019) and $7\times7\ \mathrm{km^2}$ for the short-wave infrared





(SWIR) bands (updated to $5.5 \times 7 \ km^2$ in August 2019) (Ludewig, 2021). Methane is retrieved from TROPOMI measurements
of sunlight reflected by Earth's surface and atmosphere in the SWIR.

### 2.1.1   Sentinel-5 Precursor operational XCH$_4$ product

The operational S5P methane product uses a a retrieval algorithm, which applies the physical equations describing atmospheric
radiative transfer. The algorithm simultaneously retrieves aerosol information (NIR and SWIR bands) and the methane column
(SWIR band) (and other parameters e.g. surface albedo) in order to account for the influence of aerosol scattering (Hu et al.,
2016; Hasekamp et al., 2019). The retrieval algorithm utilizes the GMTED2010 Digital Elevation Model (DEM) at a resolution
of approximately $2 \ km$ (Lorente et al., 2021) and European Centre for Medium-Range Weather Forecasts (ECMWF) data to
calculate the surface pressure from which the pressure profile is constructed. An erroneous pressure profile affects the retrieval
of XCH$_4$ twofold: through the pressure dependence of the molecular absorption cross-sections and through the retrieved air
column, which is used to convert the total CH$_4$ column to the dry air mole fraction XCH$_4$. An error of $\pm 1\%$ in the surface
pressure (ca. 10 hPa) leads to an error of roughly $1\%$ in the retrieved XCH$_4$ (ca. 20 ppb) according to Hasekamp et al. (2019).
While the operational XCH$_4$ data has proven to be of good quality by comparisons with ground-based measurements (Sha
et al., 2021) there are biases related to low- and high-albedo scenes and overestimations of XCH$_4$ over snow-covered scenes
(Lorente et al., 2021). An assessment of the operational product is also presented in Barré et al. (2021), where it is shown
that the V01 product (even after strict quality filtering) may show high bias XCH$_4$ outliers over extended regions, for example
Siberia (see their Fig. 15) related to albedo variations not fully accounted for in the retrieval (see their Fig. 16). Such features
can be easily misinterpreted as a local methane emission signal (see Froitzheim et al., 2021).
For this paper we use the versions V01.02.02, V01.03.00, V01.03.01, V01.03.02 and V01.04.00 of the S5P/TROPOMI Level
2 methane product (Apituley et al., 2017), since version V02.00.00 or newer is only available for observations after June 2021.

### 2.1.2   Sentinel-5 Precursor WFM-DOAS XCH$_4$/XCO product

The Weighting Function Modified Differential Optical Absorption Spectroscopy (WFM-DOAS) TROPOMI data product
(Schneising, 2021) is based on the WFM-DOAS algorithm (Buchwitz et al., 2006, 2007; Schneising et al., 2011, 2014), which
is a linear least-squares method based on scaling (or shifting) pre-selected atmospheric vertical profiles. The vertical columns
of the retrieved gases are calculated using the measured sun-normalized radiances by fitting a linearized radiative transfer
model to it. The scientific WFMD algorithm retrieves both methane and carbon monoxide simultaneously from TROPOMI's
SWIR bands. A detailed description of the retrieval algorithm and its differences to the operational product can be found in
Schneising et al. (2019).
The WFMD algorithm relies on accurate high resolution surface elevation information and uses GMTED2010 (Danielson and
Gesch, 2011a) with a resolution of $0.025° \times 0.025°$ as external input. As the uncertainty of the corresponding elevation data for
Greenland is larger than for other areas (see next section), potential retrieval biases may occur over Greenland. In the algorithm,
the retrieved vertical methane columns are converted into column-averaged dry air mole fractions (denoted XCH$_4$) by division





by the dry air column obtained from the ECMWF ERA5 reanalysis (Hersbach et al., 2020). In the computation of $XCH_4$, the Digital Elevation Model (DEM) is used in both the determination of the vertical methane columns and the dry air columns. In the processing of the vertical methane columns, the elevation information is required for the selection and interpolation

of suitable precalculated reference spectra. In the post-processing, the ECMWF dry air columns are corrected for the actual surface elevation of the individual TROPOMI measurements (based on the deviation from the mean altitude of the coarser model grid), inheriting the high spatial resolution of the satellite data. Therefore, errors in the elevation model led to biases in the retrieved $XCH_4$. These potential biases are mainly caused by the correction of the ECMWF dry air columns, while the impact on the retrieval of the methane columns is comparatively small. An error in the elevation data translates into an error of

the pressure, which influences the dry air column, which in turn influences the $XCH_4$. An error of 1 % in the surface pressure translates roughly to a 1 % error in $XCH_4$.

## 2.2 GMTED2010

The Global Multi-Resolution Terrain Elevation Data 2010 (GMTED2010) DEM is a suite of global terrain elevation data at three different resolutions (approximately 250, 500 and 1000 m) and different versions depending on the use case (e.g.

minimum, maximum and median elevations) (Danielson and Gesch, 2011b). While the GMTED2010 datasets provide global coverage of almost all land areas there are some exceptions. Most importantly the data for Greenland is just available in the lowest resolution (1000 m). GMTED2010 is based on various source datasets which are combined (Danielson and Gesch, 2011a). For Greenland the source data is given by Bamber et al. (2001), who developed a gridded DEM at a 1 km spacing from ERS-1 and Geosat satellite radar altimetry. The mean vertical error over the Greenland ice sheet was determined to be

$-0.33 \pm 6.97$ m while over bare rock regions it ranges from 20 to 200 m (Bamber et al., 2001). In this work we use the 30-arc-second spatial mean resolution data (Danielson and Gesch, 2011b).

## 2.3 ATLAS/ICESat-2

The Ice, Cloud and Land Elevation Satellite-2 (ICESat-2) was launched September 15, 2018. Since then, it has provided high resolution data from its sole instrument, the Advanced Topographic Laser Altimeter System (ATLAS). ATLAS uses a photon-

counting lidar and ancillary systems to measure the travel time of a photon and its geodetic latitude and longitude (Abdalati et al., 2010). The output of the single laser (532 nm) is split into six beams, which are arranged into three pairs of beams that follow three parallel Reference Pair Tracks (RPTs). The configuration allows for the measurement of the surface slope in the along- and across-track direction in a single pass. The laser footprint on the ground is approximately 17 m with a spatial sampling of 0.7 m. The middle pair is aligned to a Reference Ground Track (RGT) by the onboard software. The ICESat-2

mission gathers data along 1387 different RGTs, with a 91-day return cycle, which allows for the detection of elevation changes (Smith et al., 2021a; Neumann et al., 2019). The mission target is to reach an accuracy better or equal to $0.4$ cm $yr^{-1}$ on an annual basis over ice-sheets. The actual elevation precision depends on the signal-to-noise ratio, the length over which laser shots are accumulated, and the precision of the photon timing. The 100-shot standard deviation is estimated to be 2–9 cm over the interior ice sheet and 6–29 cm over glaciers (Neumann et al., 2019).



Here we use the ATLAS/ICESat-2 L3B Annual Land Ice Height, Version 4 (ATL11) (Smith et al., 2021b) dataset derived from the ATLAS/ICESat-2 L3A Land Ice Height product (ATL06). The ATL11 product provides height measurements, errors and quality information for a set of reference points spaced every 60 m along their corresponding ground tracks. Each of these height measurements consider ATL06 segments whose centers lie within 60 m along-track and 65 m across-track of a reference point. The data span 29 March 2019 to 23 June 2021 providing nine measurements per reference point which are 91 days apart.

## 3 Methods

In our analysis we use a grid resolution of $0.1° \times 0.2°$ for both the elevation data and the methane data. At 60°N this corresponds to a resolution of roughly $11 \times 11 \text{ km}^2$, which is slightly larger than the resolution of S5P at $7 \times 7 \text{ km}^2$. We also define two regions of interest which will be the focus of our investigation (see Fig. 1); the regions were chosen due to the strong height differences between GMTED2010 and ICESat-2 data in these areas (see Fig. 3). Region one lies on the north-west coast of 130 Greenland (74°N, 64°W, 77°N, 54°W) and region two is located on the eastern coast (65°N, 40°W, 70°N, 20°W).

### 3.1 Seven-day methane anomaly

To account for the seasonal variability of methane and the overall increase of methane concentrations (AMAP, 2015) we calculate the average of 7-day $XCH_4$ anomalies (Fig. 2). We define the 7-day $XCH_4$ anomaly as follows: First we calculate the daily mean $XCH_4$ for every gridcell. In the best case this returns a daily time series of methane measurements for every 135 grid cell. However this time series will have gaps (e.g. measurements not passing the quality filter). To account for this we average the $XCH_4$ time series over 7-days (i.e. we have a methane measurement every 7-days for all cells in the ideal case). In the last step we calculate the methane anomaly by calculating the reference methane, which is defined by the average in the reference area (73°N, 48°W, 78°N, 38°W), and subtracting it from the 7-day methane averages. This yields the 7-day $XCH_4$ anomaly. We chose the reference area due to three factors: proximity to the observed regions, absence of methane sources and 140 data coverage. The $XCH_4$ in the reference area is typically lower than in the coastal regions due to the higher elevation (see Sec. 4.3). Since the elevation stays virtually constant in the relevant time frame, this effect merely shifts the reference point of the anomaly. The seven-day methane anomaly is denoted by $\Delta XCH_4$.

### 3.2 Processing of ICESat-2 data

We use the ATL11 (Smith et al., 2021b) data as a basis for our elevation grid. The data is split into three groups for each RPT. 145 The data along each RPT $p(x,t)$ is ordered by positions $x$ along the track for which up to nine observations for different return cycles, $t$, are available. We iterate over all positions $x$ for each RPT and get the array of return cycle observations $p_x(t)$. We remove NaN's and observations with a quality flag ($qa > 0.1$) and use the remaining data $\tilde{p}_x(t)$ for the gridding process. Next we calculate the total error for each measurement in $\tilde{p}_x(t)$ according to Smith et al. (2021a) and calculate the average of the height measurements using the total errors as weights. The weighted average height then gets added to the corresponding grid 150 cell. After all data has been processed, each grid cell containing the sum of all collocated heights is divided by the number of





collocated data points.

We also mention the recent publication of Fan et al. (2021), which introduces a new DEM for Greenland generated from ICESat-2 data. Since the publication by Fan et al. (2021) was still in review when we conducted the main part of our work we used our own gridded ICESat-2 data. In Appendix A we present a comparison of our gridded data with this new DEM and

show that the differences don't effect the results and conclusions of our work.

## 4    Results

### 4.1    Methane anomaly

In Fig. 1 we present the mean $XCH_4$ between 2018–2020 for the operational and the WFMD product. The operational product shows distinct areas of low/high methane concentration on the edges of Greenland. For the WFMD product mainly regions

of high concentration are visible on the edges of Greenland, with the exception of region two, which shows an area of low concentration. The effect of the elevation on $XCH_4$, explained below in Sec. 4.3, can also be seen over the ice shield for both products (i.e. decreasing $XCH_4$ with increasing ground height). Figure 2 shows the mean 7-day methane anomaly ($\Delta XCH_4$) for both products. Here we can observe the same features as in Fig. 1.

### 165    4.2    Comparison of elevation data

To compare GMTED2010 with the gridded ICESat-2 data we resample it to the same $0.1° \times 0.2°$ grid using cubic resampling. In Fig. 3 we show the height difference $\Delta H$ between both elevation data. On the north-western coast (region one) we see a region of positive elevation differences of roughly 100–200 m. This corresponds to regions of elevation change reported by Shepherd et al. (2020). On the south-eastern coast (region two) we observe a distinct feature consisting of neighboring positive

and negative elevation differences.

### 4.3    Height correction

Since Greenland is a region with very large elevation differences we have to account for the influence of the terrain height on the retrieved $XCH_4$. This is due to the elevation-dependent weighting of tropospheric and stratospheric contributions to $XCH_4$ which leads to decreasing $XCH_4$ with increasing height. In Fig. 4 and Fig. 5 we show the correlation between the terrain height

used in the retrieval and the $XCH_4$ for the WFMD data and the operational data respectively. For both cases we see a downward trend of $XCH_4$ with increasing height. We calculate a linear fit for both cases and use the slope as a linear correction factor in our plots (denoted as 'height corrected').





### 4.4 Correlation between $\Delta H$ and $\Delta XCH_4$

In this section we present the correlations between $\Delta H$ and $\Delta XCH_4$ to investigate and quantify how errors in the topography

data lead to a change of the $\Delta XCH_4$. In Fig. 6 we show the correlation between $\Delta H$ and $\Delta XCH_4$ for the WFMD product. In the top row we show the correlation of the raw $\Delta XCH_4$ data and in the bottom row we show the correlation for the height corrected case (see Sec. 4.3 for an explanation). We observe a linear correlation with $p > 0.8$ (Pearson correlation coefficient) for the corrected data. Figure 7 shows the same correlation plots but for the operational data product. The correlation coefficients are smaller than for the WFMD product; for region one the correlation factor becomes smaller after height correction.

In contrast to the WFMD data, both regions show a large scatter of $\Delta XCH_4$ around $\Delta H \approx 0$ obfuscating the linear relationship between $\Delta H$ and $\Delta XCH_4$.

In Fig. 8 and Fig. 9 we show the correlation plots for all of Greenland for the WFMD and the operational product respectively. For the WFMD product the correlation is again improved by the height correction ($p = 0.49$). For the operational data we observe a greater spread of values and a less significant correlation with $p = 0.27$ in the height corrected case.


### 4.5 Updated WFMD product

Finally, we present a preliminary version of an updated WFMD product which uses the Greenland DEM from Fan et al. (2021) instead of GMTED2010. Figure 10 shows the WFMD v1.5 and the improved version with the more accurate DEM next to each other. The dominant features in regions one and two are no longer visible in the updated version. In addition, smaller

features along the whole coastline of Greenland is no longer visible. A difference between both versions is shown in Fig. 11. The areas of great difference (e.g. region 1 and 2) correspond nicely to the differences observed between GMTED2010 and ICESat-2 data in Fig. 3. Even though there is no validation with third party products, the updates to the WFMD product create an overall smoother and more realistic methane distribution and thus present significant improvements of the WFMD product over Greenland.

## 5    Conclusions

In this study we investigated the presence of strong methane anomalies along Greenland's coastline; both for the operational S5P methane product and the WFMD product (see Fig. 1 and Fig. 2). Our hypothesis was that some of these anomalies can be explained by the use of inadequate topography data from GMTED2010 in both algorithms. We tested this hypothesis by calculating the height difference $\Delta H$ between GMTED2010 and elevation data from ICESat-2 and correlating it with the mean

7-day methane anomaly $\Delta XCH_4$. For the WFMD product we observed good correlations for region one and two (Fig. 6) as well as all of Greenland (Fig. 8). The correlation factors where improved by accounting for the underlying height relationship due to the elevation dependent weighting of stratospheric air using the linear correction factor of $14.8 \; \mathrm{ppb \; km^{-1}}$ (see Sec. 4.4). Finally, we present an updated version of the WFMD product in Fig. 10 which uses updated elevation data from Fan et al.





(2021); here we can see that features discussed in this paper disappeared, thus demonstrating that they were caused by the
usage of outdated topography data.

For the operational S5P methane product the correlations were less clear (Fig. 7 and Fig. 9). While the correlations for region
one as well as the whole Greenland region have high uncertainties, the correlations for region 2 have smaller uncertainties
and are thus more clearly identifiable but have significant noise. We argue that the same effect seen in the WFMD data can
be observed for the operational data. However multiple factors obfuscate this effect. Firstly, the coverage of the operational
product is lower in the investigated regions (see Fig. 2), this data gap includes bare rock regions which show large uncertainties
in the GMTED2010 data (see Sec. 2.2). Thus, part of the areas responsible for the correlations are missing in this product.
Secondly additional effects of higher magnitude are present in the operational data. This can be seen in Fig. 5 which shows
large methane anomalies, which are not related to height anomalies. While the source of these effects is not known, possible
candidates are albedo-related biases or issues in snow-covered scenes previously described by Lorente et al. (2021).

Our investigations show that the use of wrong or inadequate topography information introduces significant bias on the S5P
$XCH_4$ data products at around $\pm 50 - 100$ ppb. This issue can be resolved by using adequate topography data where available.
Ideally the topography would be regularly updated as the elevation of especially glaciated regions continues to change. For the
WFMD product a new version with updated topography data for Greenland is currently being prepared. Lorente et al. (2021)
proposed the usage of Shuttle Radar Topography Mission (SRTM) data instead of GMTED2010 to be incorporated into the
next processor update for the S5P operational methane product. While the higher spatial resolution data may help, the SRTM
data is from the year 2000 and thus cannot capture the changes to Greenland's ice shield which occurred during the last twenty
years. In App. B we discuss the seven-day methane anomaly for the scientific SRON product (which uses SRTM data) which
shows improvements in comparison to the operational data product. In conclusion, we recommend the use of the most precise
and appropriately timed DEM available.

We have demonstrated in this study that iterative testing and investigation prove to be vital to ensure the quality of S5P methane
data products. While the focus of this paper was on Greenland, we note that biases due to inadequate or inaccurate elevation
data may also arise in other regions. This is especially true for polar regions, because the topography can change over a few
years (e.g. glacier ice loss, see Willis et al. (2018)) and DEM are rarely updated. While the spatial extent of problematic regions
outside of Greenland is probably much smaller and DEM-related biases are typically expected to be smaller we still expect
significant biases in other areas where either DEMs have high inaccuracies or notable change in the topography occurred since
the creation of the DEM. We want to emphasize that both effects are important to consider for present S5P data products as
well as for future missions that need DEM as input data.

*Code and data availability.* TROPOMI scientific WFMD methane data product available at https://www.iup.uni-bremen.de/carbon_ghg/
products/tropomi_wfmd/. TROPOMI operational methane data product available from https://sentinels.copernicus.eu/web/sentinel/missions/
sentinel-5p/data-products. TROPOMI scientific SRON data product available at https://ftp.sron.nl/open-access-data-2/TROPOMI/tropomi/



ch4/18_17/. GMTED2010 data available at https://earthexplorer.usgs.gov/. ATL11/ICE-Sat2 data available at https://nsidc.org/data/ATL11/versions/4. Code available per request from the author.

## Appendix A:  Difference between gridded ICESat-2 data and new Greenland DEM

In our analysis we use gridded ATL11/ICESat-2 data as described in Sec. 3.2. The updated WFMD product we use for comparison is however based on a recently published Greenland DEM based on ATL11/ICESat-2 data (Fan et al., 2021). In Fig. A1 we show the difference between our data and the downsampled DEM by Fan et al. (2021). Most differences seem to occur over bare rock regions at the edges of Greenland. Here we can observe differences of up to 200–300 m. We assume this happens because we use a very simple gridding method without any additional filtering criteria (except the quality flag). Thus we redid

part of our analysis with the Fan et al. (2021) DEM which can be seen in Fig. A2. Comparisons between Fig. 6 and Fig. A2 show that the differences are small. Use of the new DEM changes the correlation factors by roughly 0.02 in the height corrected case. Since the differences are small and don't change the interpretation of our results we conclude that the use of our own gridded ICESat-2 data introduces no significant errors to our analysis. Figure A3 shows the updated version of Fig 7.

## Appendix B:  Seven-day methane anomaly of the scientific SRON XCH$_4$ product

While our analysis focused on the WFMD and operational data products we want to mention improvements made in the scientific SRON product compared to the operational data, showing a similar smooth and realistic methane distribution as the updated WFMD data product. Figure A4 shows the seven-day methane anomaly for the operational and scientific data products. The strong negative $\Delta$XCH$_4$ anomalies at the edge of Greenland vanish for the scientific product, which we identify with improvements made at low surface albedo scenes (Lorente et al., 2021). Strong positive anomalies are reduced for region

two but stay present for other coastal regions of Greenland. We assume that the use of SRTM elevation data (Lorente et al., 2021) lead to improvements in region two, due to the higher resolution in comparison to GMTED2010. In other areas changes in the elevation (e.g. due melting of glaciers) are not captured by SRTM data which is around 20 years old. Therefore we recommend to use an updated topography in future releases.

*Author contributions.* JH produced the methods and the results. OS provided a preliminary version of the WFMD product used in the

analysis. JH wrote the original draft with input from OS, MiB and MaB. All authors contributed to the final version of the paper.

*Competing interests.* The authors declare that they have no conflict of interest. Several co-authors are co-editors of AMT.



*Acknowledgements.* This research is funded by the University of Bremen, as part of the junior research group 'Greenhouse gases in the Arctic'. We gratefully acknowledge the funding by the Deutsche Forschungsgemeinschaft (DFG, German Research Foundation) – Projektnummer 268020496 – TRR 172, within the Transregional Collaborative Research Center "ArctiC Amplification: Climate Relevant Atmospheric

and SurfaCe Processes, and Feedback Mechanisms (AC)³". This research also received funding from the European Space Agency (ESA) Climate Change Inititative (CCI) via project GHG-CCI+ (ESA contract No. 4000126450/19/I-NB). This publication contains modified Copernicus Sentinel data (2018-2020). Sentinel-5 Precursor is an ESA mission implemented on behalf of the European Commission. The TROPOMI payload is a joint development by the ESA and the Netherlands Space Office (NSO). The Sentinel-5 Precursor ground-segment development has been funded by the ESA and with national contributions from the Netherlands, Germany, and Belgium. The TROPOMI/WFMD retrievals

presented here were performed on HPC facilities of the IUP, University of Bremen, funded under DFG/FUGG grant INST 144/379-1 and INST 144/493-1. The pre-operational TROPOMI data processing was carried out on the Dutch national e-infrastructure with the support of SURF Cooperative. Scientific color maps (Crameri, 2021) are used in this study to prevent visual distortion of data and exclusion of readers with color-vision deficiencies (Crameri et al., 2020).



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

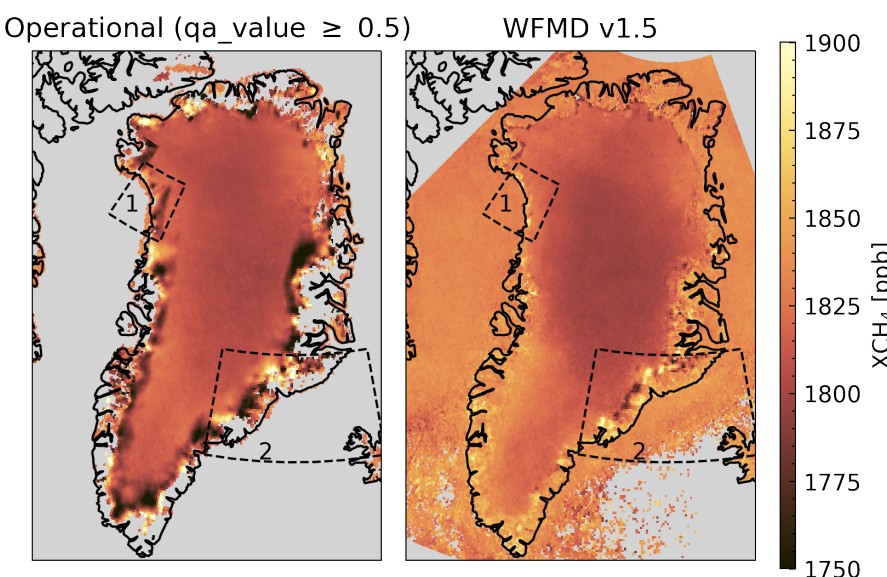

**Figure 1.** Comparison of the mean $XCH_4$ from 2018-2020 over Greenland for the operational methane data product (left) and the WFMD product (right). The dashed boxes show areas investigated in this work.

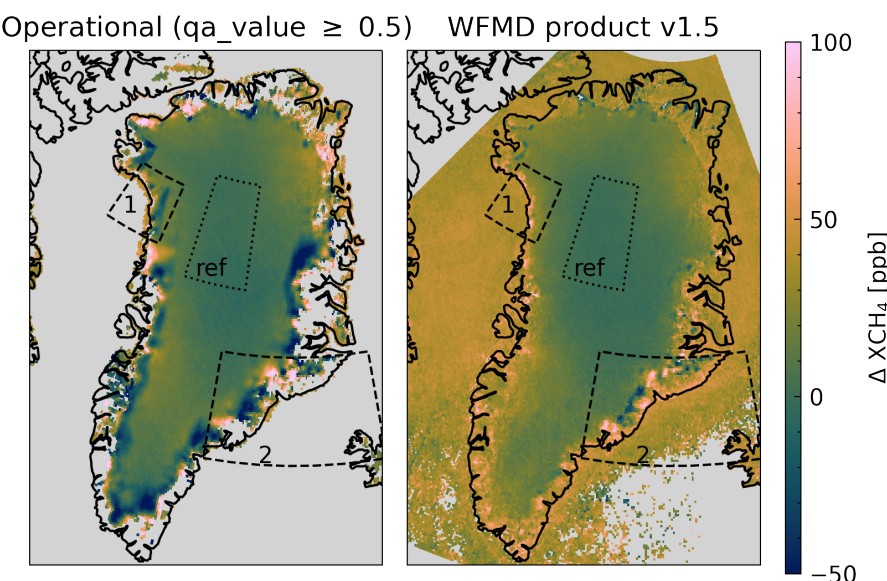

**Figure 2.** Comparison of the mean 7-day XCH$_4$ anomaly from 2018-2020 over Greenland in the operational methane data product (left) and the WFMD product (right). The reference area is shown by the dotted contour. The dashed boxes show areas investigated in this work.

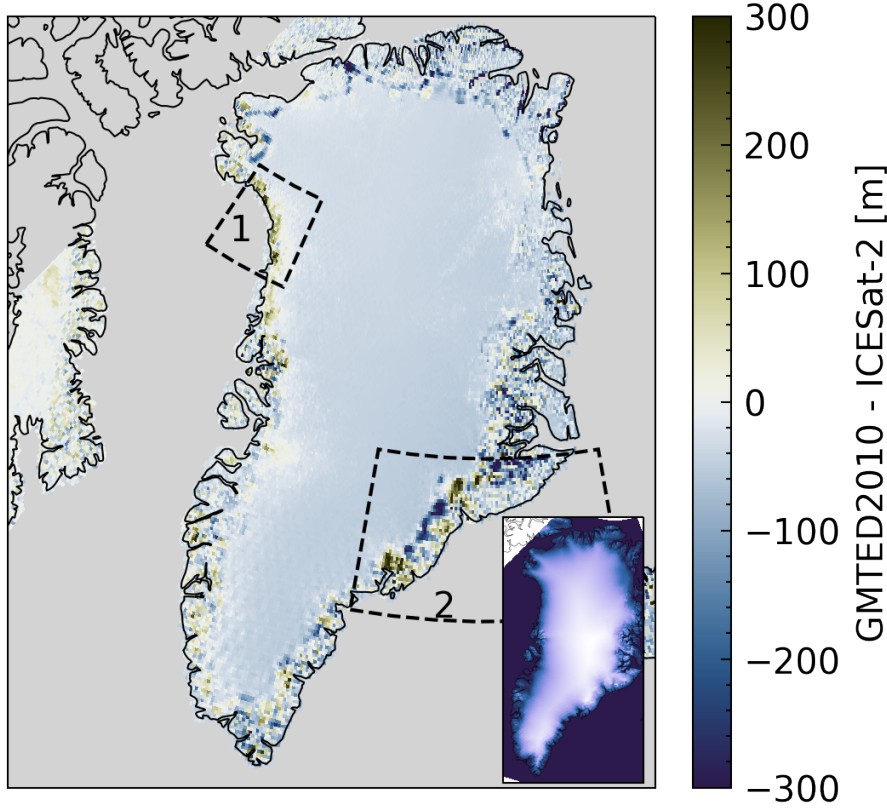

**Figure 3.** Height difference between GMTED2010 and gridded ICESat-2 data. Inset shows Greenland topography according to GMTED2010. Large height differences can be observed in regions one and two. Smaller differences can be seen on the whole Greenland coastline.



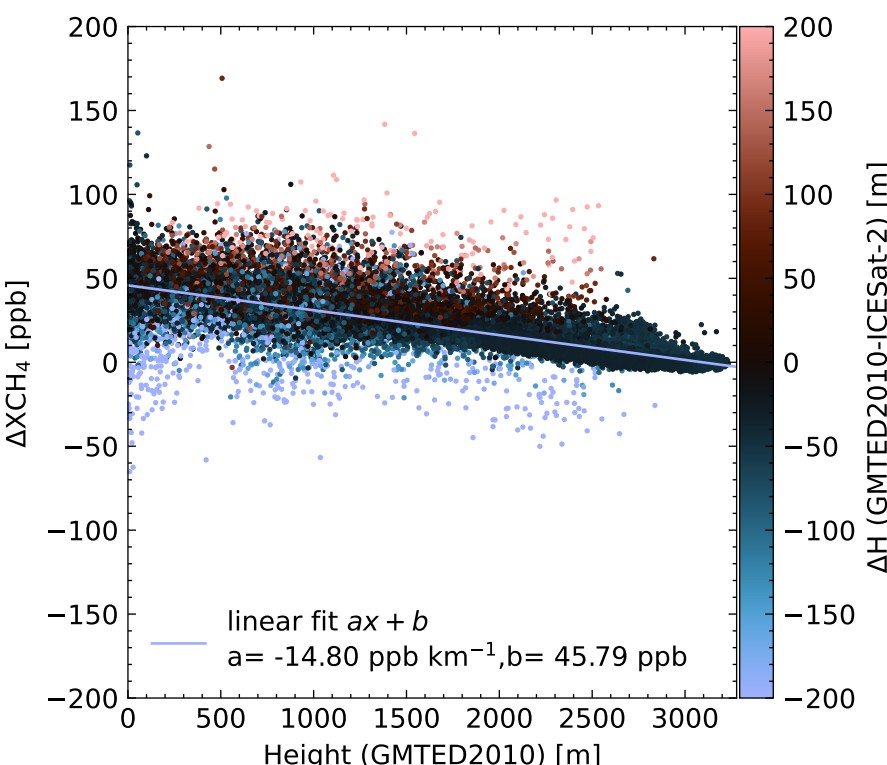

**Figure 4.** Correlation between height from GMTED2010 and the mean 7-day methane anomaly for the WFMD product. The slope of the linear fit is used in the height correction.



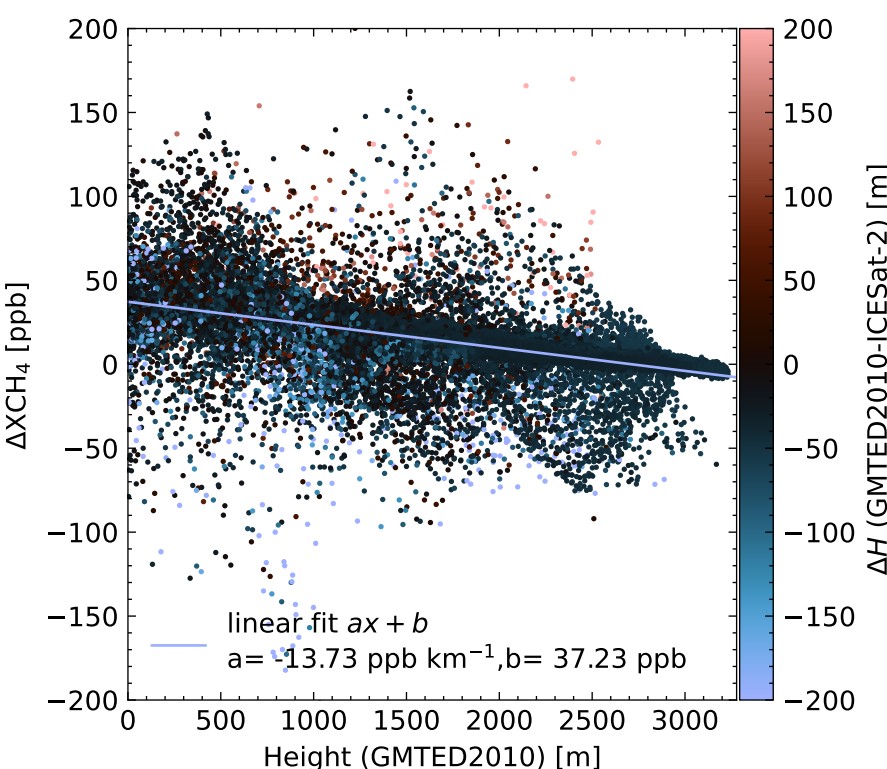

**Figure 5.** Correlation between height from GMTED2010 and the mean 7-day methane anomaly for the operational product. The slope of the linear fit is used in the height correction.

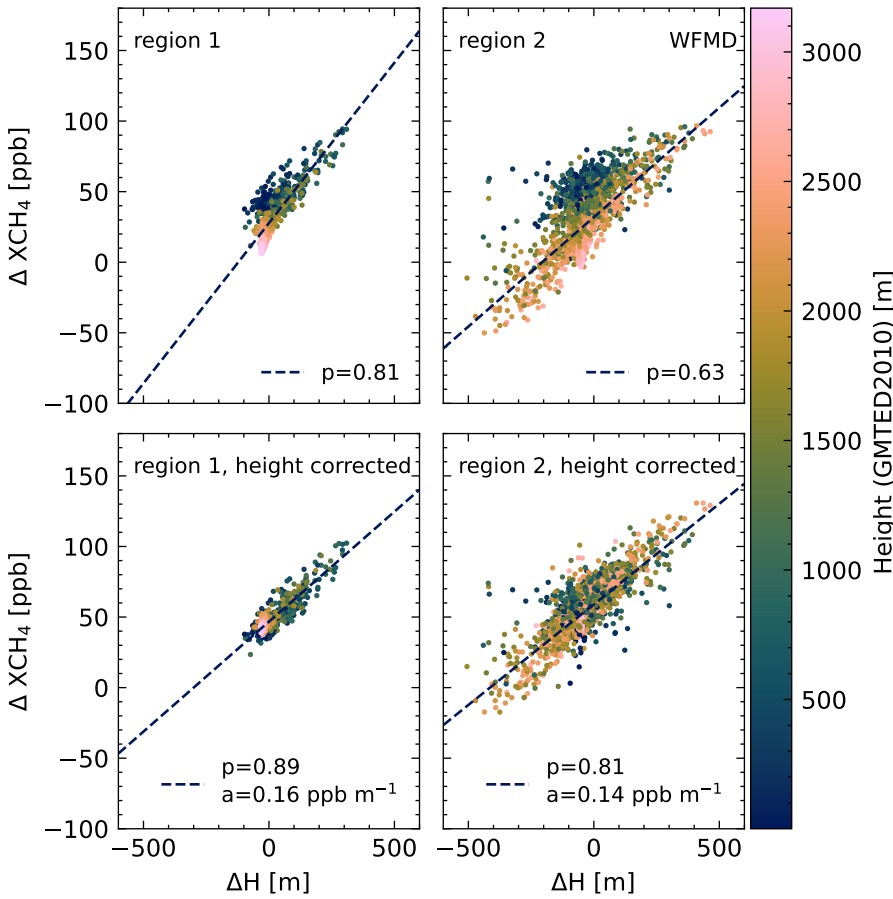

**Figure 6.** Correlation between mean 7-day methane anomaly $\Delta XCH_4$ and $\Delta H$ for the WFMD product. In addition to the effect of $\Delta H$ on the $XCH_4$, the apparent elevation has a linear effect on the retrieved $XCH_4$. This is due to the elevation-dependent weighting of tropospheric and stratospheric contributions to $XCH_4$. We thus see lower $\Delta XCH_4$ for measurements over the high ice shield (see upper left panel). This height relationship is corrected in the lower panels using a linear correction factor (see Fig. 4).



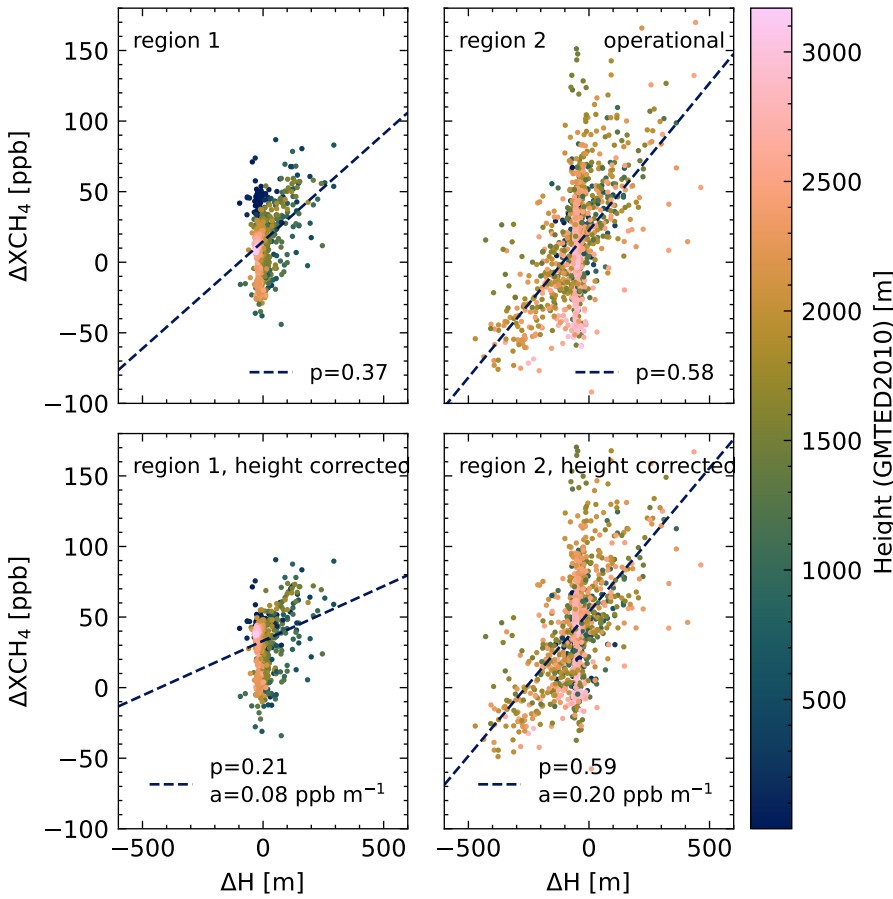

**Figure 7.** Correlation between mean 7-day methane anomaly $\Delta XCH_4$ and $\Delta H$ for the operational product. In addition to the effect of $\Delta H$ on the $XCH_4$, the apparent elevation has a linear effect on the retrieved $XCH_4$. This is due to the elevation-dependent weighting of tropospheric and stratospheric contributions to $XCH_4$. We thus see lower $\Delta XCH_4$ for measurements over the high ice shield (see upper left panel). This height relationship is corrected in the lower panels using a linear correction factor (see Fig. 5).

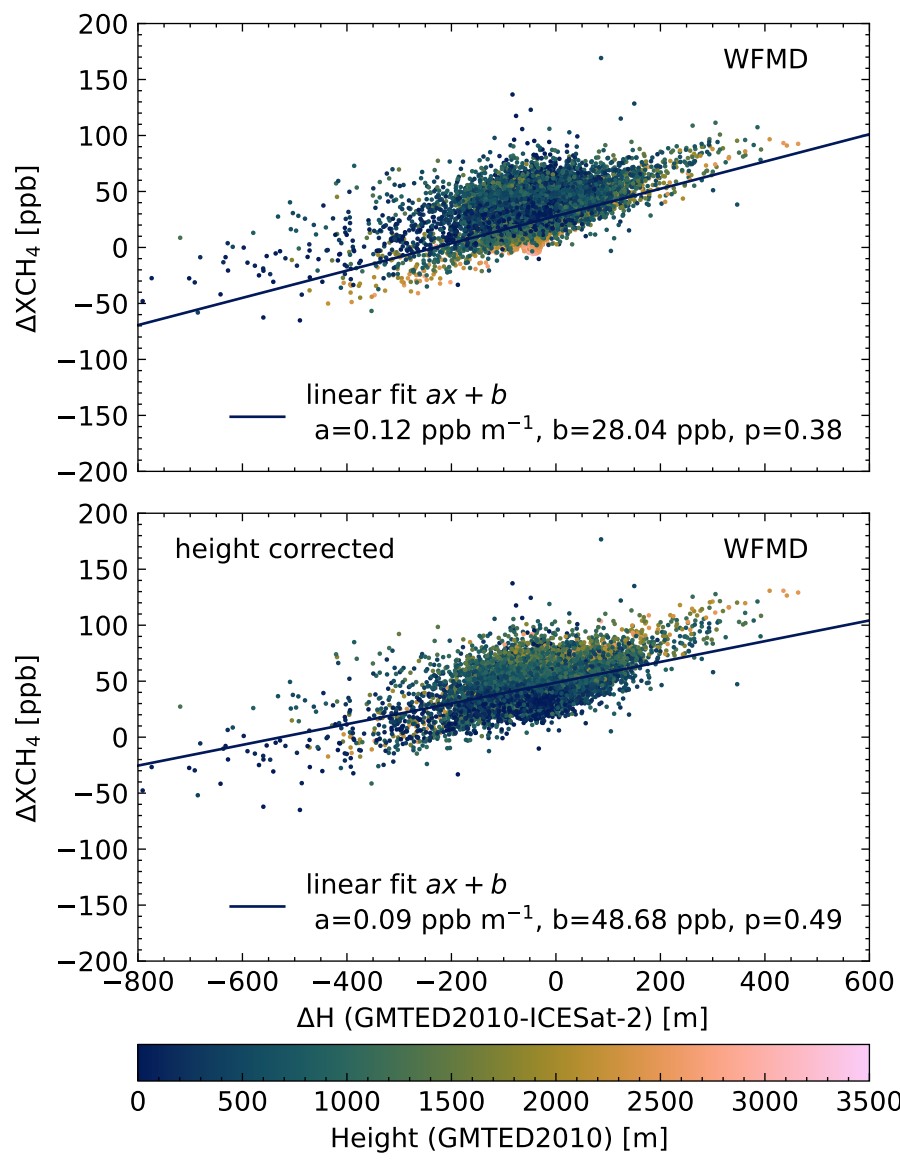

**Figure 8.** Correlation between mean 7-day methane anomaly and ΔH for WFMD product (whole Greenland region). See Fig. 6 for an explanation of the height correction.

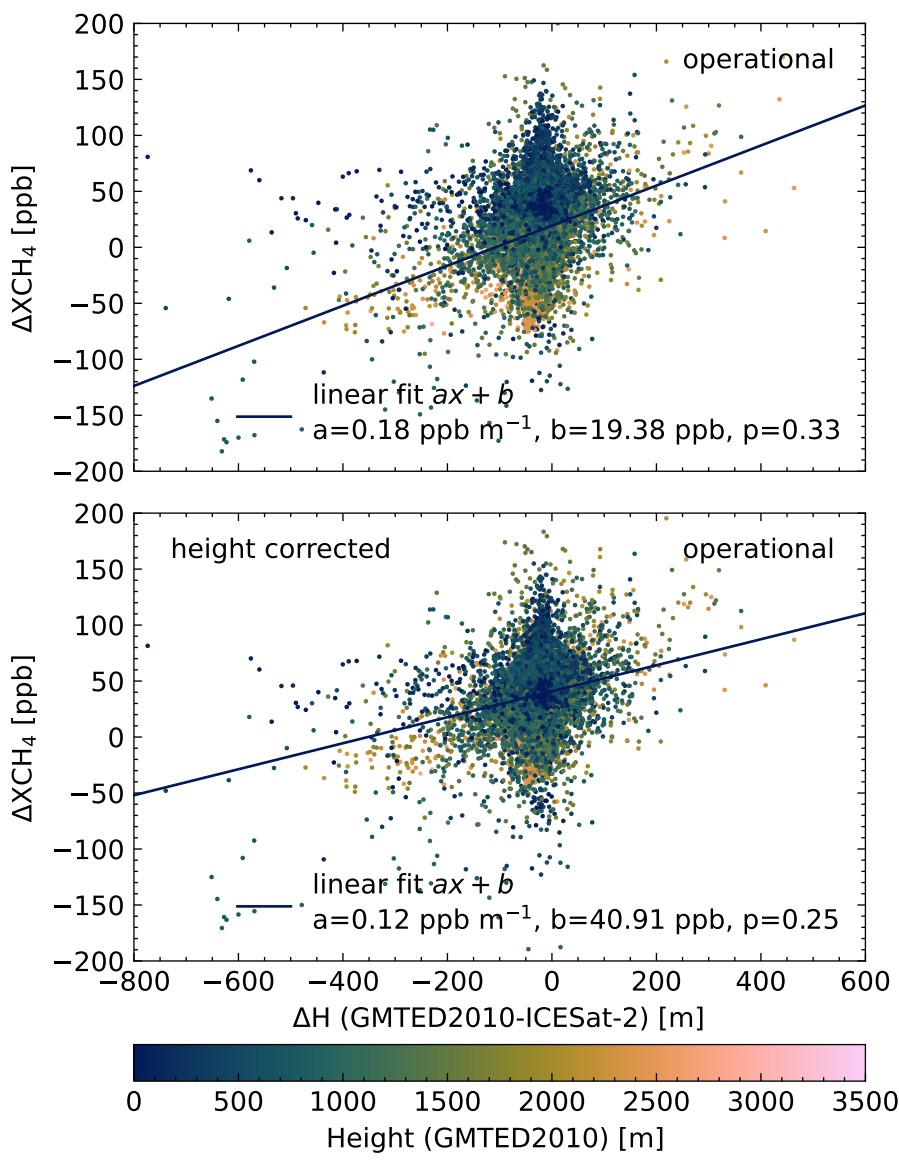

**Figure 9.** Correlation between mean 7-day methane anomaly and ΔH for operational S5P product (whole Greenland region). See Fig. 7 for an explanation of the height correction.

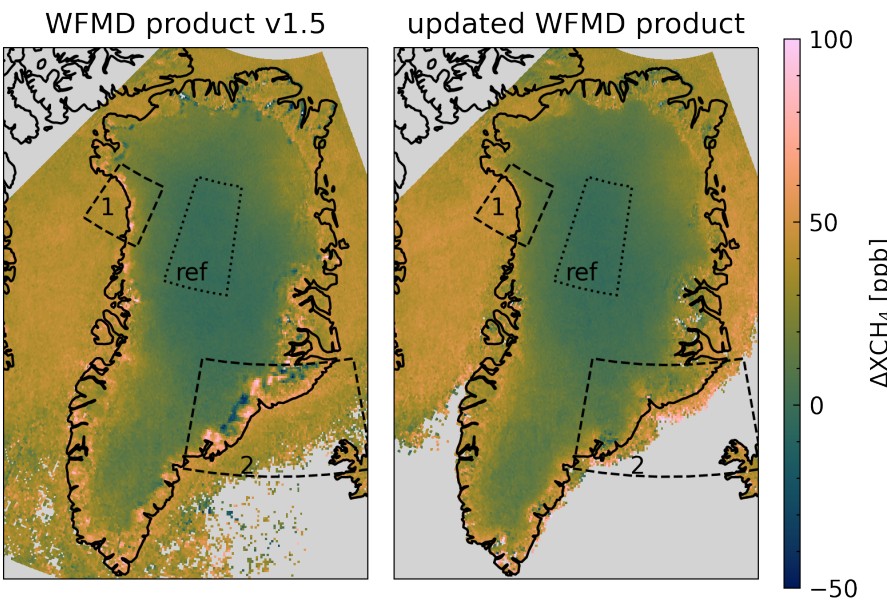

**Figure 10.** Comparison of the mean 7-day XCH$_4$ anomaly from 2018-2020 over Greenland in the WFMD v1.5 product (left) and an updated version of the WFMD product (right). The distinct features in region one and two vanish for the updated WFMD product. The reference area is shown by the dotted contour. The dashed boxes show areas investigated in this work. The lower coverage over the ocean for the updated product is due to stricter quality filter criteria.



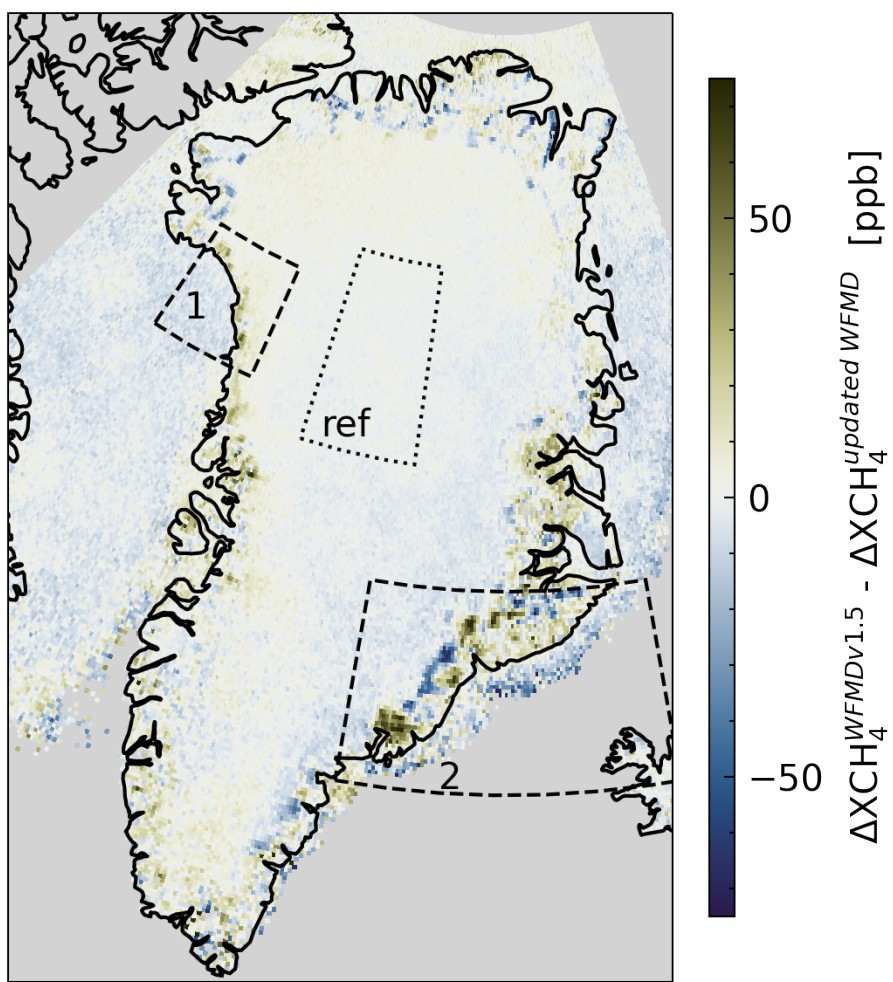

**Figure 11.** Difference of the mean 7-day $XCH_4$ anomaly from 2018-2020 over Greenland between WFMD v1.5 and the updated WFMD product. The reference area is shown by the dotted contour. The dashed boxes show areas investigated in this work. The lower coverage over the ocean for the updated product is due to stricter quality filter criteria.



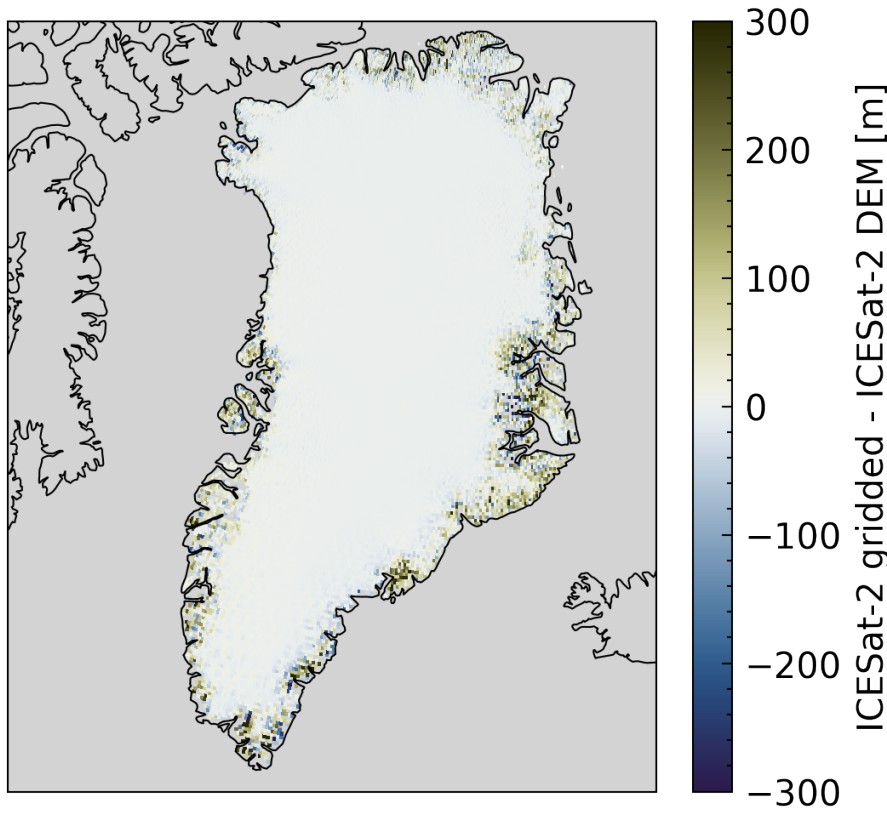

**Figure A1.** Difference between own gridded ICESat-2 data and DEM from ICESat-2 data by Fan et al. (2021). Differences mainly occur on the edges of Greenland where height errors can be high over rocky regions.



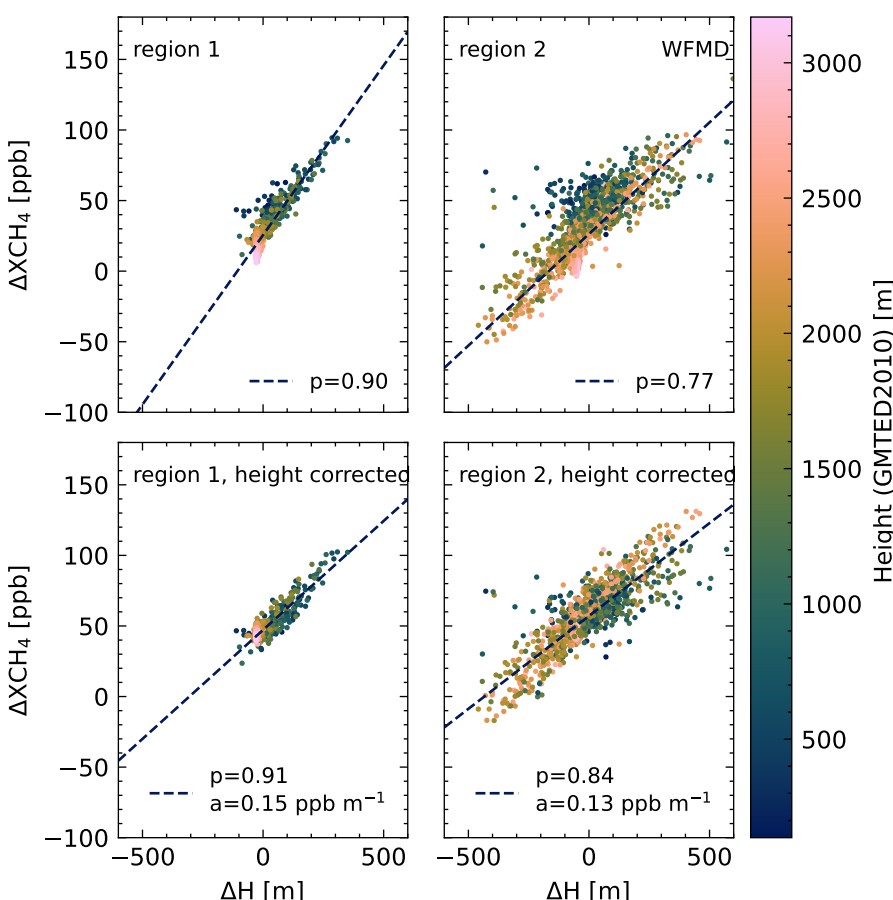

**Figure A2.** Correlation between mean 7-day methane anomaly $\Delta XCH_4$ and $\Delta H$ for WFMD product. Here we use the DEM from Fan et al. (2021) instead of own gridded ICESat-2 data. See Fig. 6 for an explanation of the height correction.



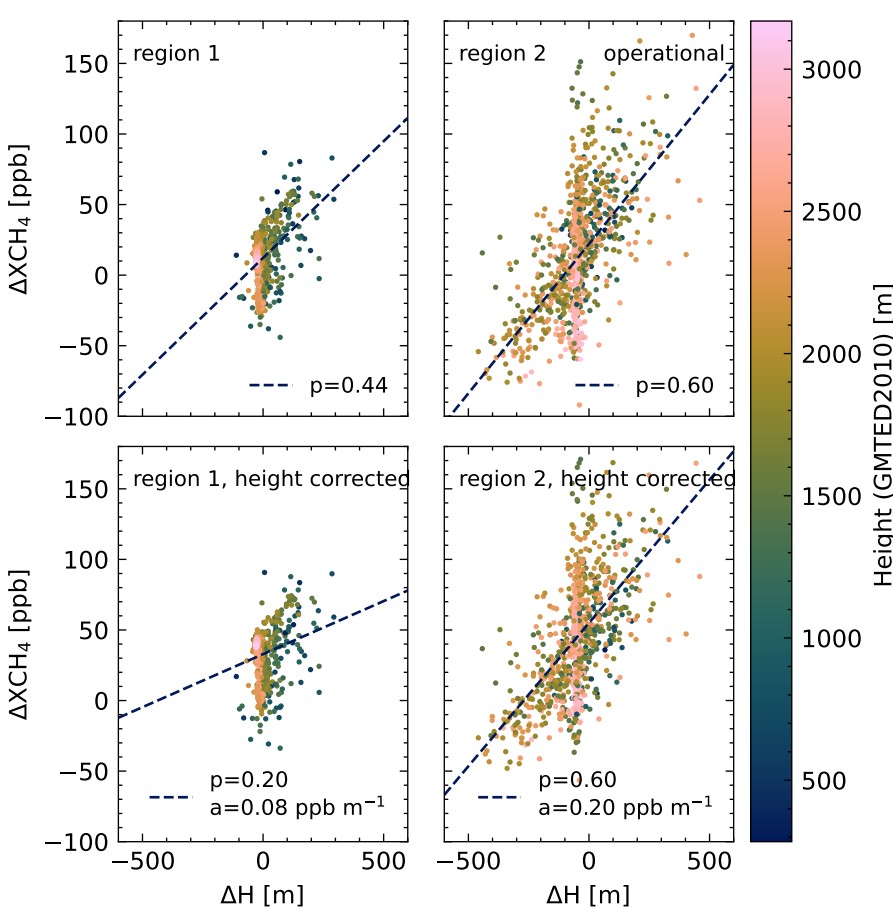

**Figure A3.** Correlation between mean 7-day methane anomaly $\Delta XCH_4$ and $\Delta H$ for operational product. Here we use the DEM from Fan et al. (2021) instead of own gridded ICESat-2 data. See Fig. 7 for an explanation of the height correction.



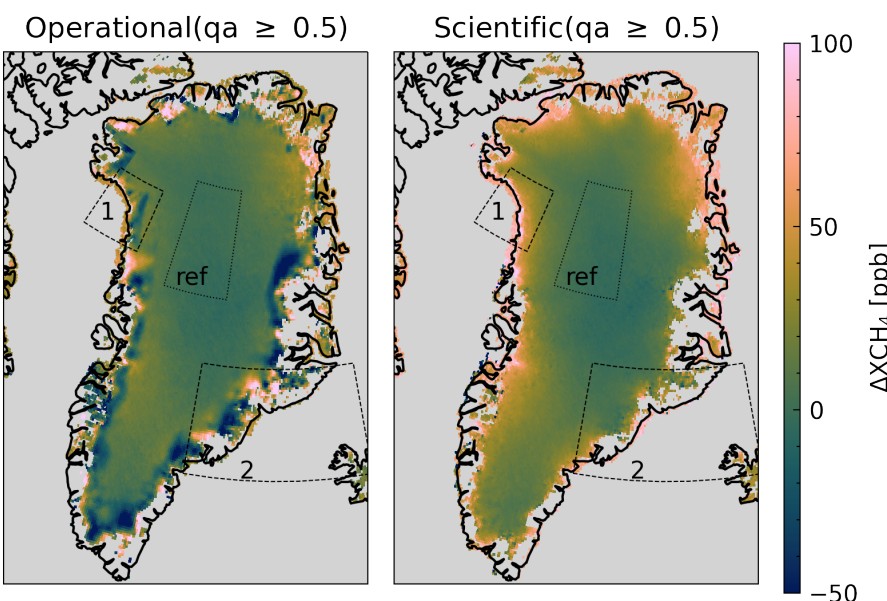

**Figure A4.** Comparison of the mean 7-day XCH$_4$ anomaly from 2018-2020 over Greenland in the operational product (left) and the scientific SRON product (right). The reference area is shown by the dotted contour. The dashed boxes show areas investigated in this work.