# Peer review of "On the influence of underlying elevation data on Sentinel-5 Precursor satellite methane retrievals over Greenland"

_Atmospheric Measurement Techniques, 2022_

## Author Comment (AC1)

**Reply to reviewer 1**

Jonas Hachmeister

June 10, 2022

We would like to thank the reviewer for their helpful comments. Below the specific comments are listed in **bold text**. Our replies are in standard black text and the changed passages from the paper are in blue.

**L. 52-54: Why give the old spatial resolution upfront and current resolution parenthetically? Consider reversing to give current specs first.**

The order was reversed in the final manuscript.

**L. 61: Recommend defining the GMTED2010 acronym here.**

We added the definition of the GMTED2010 acronym to the introduction.

In this paper we investigate noticeable features in the maps of retrieved $XCH_4$ over Greenland which can be seen both in the operational S5P $XCH_4$ product and the S5P WFMD product. For this we investigate the digital elevation model (DEM) used in both retrievals, namely the Global Multi-resolution Terrain Elevation Data (GMTED2010) and compare it to new elevation data from the ICESat-2 satellite mission.

**L. 76: You use WFM-DOAS here but WFMD elsewhere.**

WFM-DOAS will be changed to WFMD in the whole text.

**L. 93-96: Bias of a few percent could be significant for inference of regional emissions, no?**

This is true and is one of the reasons for aiming to reduce the bias originating from the inaccurate DEM.

**L. 135: Recommend mentioning/defining the TROPOMI quality filter somewhere in Section 2.**

We use the quality flags as defined by the product user guides of the operational and WFMD product. We added information on the quality screening to both sections.

Section 2.1.1:

The product includes a quality assurance value (qa) which is a continuous quality descriptor ranging from 0 (no data) to 1 (full quality data). As recommended in the product user guide [citation] we exclude data with qa$< 0.5$.

Section 2.1.2:

In another post-processing step the data is quality filtered using a machine learning approach based on a random forest classifier [citation]. We use data with a quality flag qf$= 0$ (good) and don't include data with qf$= 1$ (potentially bad).

**L. 141-142: What do you mean by "this effect merely shifts the reference point of the anomaly". As written it's not clear what the "effect" is. I think you mean that the selection of reference area defines the reference point for the anomaly, is that right?**

Yes, the selection of the reference area defines the reference point for the anomaly. We changed the sentence to be more clear.

Since the elevation stays virtually constant in the relevant time frame, the choice of the reference area merely defines the reference for the anomaly. The seven-day methane anomaly is denoted by $\Delta XCH_4$.

**L. 147: Wouldn't removing observations with quality flag $> 0.1$ remove most data? Should it not be $< 0.1$? (I'm not familiar with ICESat-2 data conventions.)**

According to the IceSat-2 ATBD quality flag$= 0$ (qf) is used for the highest quality data. Data with qf$= 1$ doesn't fulfill all the quality criteria (see product user guide of ATL11 data). In the code we used the condition qf$> 0.1$ which is of course identical to the condition qf$= 1$ since the qf is either 0 or 1. We clarified this in the updated version of the manuscript:

We remove NaN's and only include observations with quality flag qf$= 0$, meaning high quality data...

**L. 149: Are these weights inversely proportional to the errors? Using the errors as weights directly seems like it would more strongly weight higher-uncertainty data.**

The inverse errors are used as weights. This is clarified in the updated manuscript.

Next we calculate the total error for each measurement in $\tilde{p}_x(t)$ according to [citation] and calculate the average of the height measurements using the inverse total errors as weights.

**L. 150: Consider adding an equation here to unambiguously describe the approach.**

We do not consider it important to include the equations, since the precise gridding method is not relevant for the results of this paper. For example one can use either the mean or the median of all points in a grid cell with almost identical results. Furthermore we showed that the differences between our gridded data and the Greenland DEM also based on ICESat-2 data (which uses a more sophisticated gridding method) (see Appendix A) is not relevant for the conclusion of our paper.

**L. 168-169: This phrasing seems to imply that the differences between ICESat-2 and GMTED2010 are due to ice sheet dynamics, but isn't 100-200 m too extreme for that to be the case? I don't know much about ice sheets, but from Section 2 the error seems mostly related to the radar altimetry.**

We clarified the phrasing. We believe that both effects are present in Greenland with different contributions for different regions. However an attribution to these error sources is out of the scope of this paper. Additionally the difference in resolution of both data might play a role.

To compare GMTED2010 with the gridded ICESat-2 data we resample it to the same $0.1° \times 0.2°$ grid using cubic resampling. In Fig. 3 we show the height difference $\Delta H$ between both elevation data. On the north-western coast (region one) we see a region of positive elevation differences of roughly 100–200 m. This corresponds to regions of elevation change reported by [citation], which could explain part of this difference. On the south-eastern coast (region two) we observe a distinct feature consisting of neighboring positive and negative elevation differences. In both cases we assume that large uncertainties in the GMTED2010 data (see 2.2) in combination with local ice sheet loss and/or movement and difference in resolution of both datasets is responsible for the observed differences.

**L. 182: Recommend using "r" or "rho" for the Pearson correlation coefficient, not "p", because "p" can easily be mistaken for the p-value of the regression. This confused me on my initial review of the figures.**

This is changed in the updated manuscript.

**Section 4.5: Is there a reason not to show maps of height-corrected WFMD v1.5 $\Delta$XCH4 (or XCH4) and height-corrected operational $\Delta$XCH4 (or XCH4)? Perhaps these could be added to Fig. 10 or Fig. 12, or made into a separate new Figure. I understand the paper already has quite a few figures and you show how the scatter plots improve from the linear height correction – but I was surprised not to see how the final methane maps improve post-correction.**

The height correction is used to account for the underlying actual height dependency of methane and to thus improve the correlations between the height difference $\Delta H$ and the methane anomaly $\Delta XCH_4$. This improvement can be best seen in the already included scatter plots. One could also create methane maps using the height-correction, however this is out of scope for this work. The correction is not applied to the final data sets in general, it is just used here for the correlation analysis to expose the DEM-related errors. The height dependency of $XCH_4$ is a real effect and not an artifact like the features we investigated and that stem from the erroneous DEM. To make this difference (and the need to disentangle these two effects in the correlation analysis) clearer, we have revised Section 4.3 accordingly:

Greenland is a region with very large elevation differences. We have to account for the actual influence of the terrain height on $XCH_4$ due to the elevation-dependent weighting of tropospheric and stratospheric air. We aim to identify potential artifacts in the retrieved $XCH_4$ due to DEM inaccuracies. This real impact of topography leads to decreasing $XCH_4$ with increasing height. In Fig. 8 and Fig. 9 we show the corresponding correlation between the terrain height used in the retrieval and the $XCH_4$ for the WFMD data and the operational data respectively. For both cases, as expected, we see a downward trend of $XCH_4$ with increasing height. We calculate a linear fit for both cases and use the slope as a linear correction factor in our plots (denoted as 'height corrected'). This allows a more conclusive correlation analysis between $\Delta H$ and $\Delta XCH_4$ after disentangling the described actual altitude dependency.

**L. 210 & 222-229: Going back to my question about ice sheet changes over time – can you say more about what causes the GMTED2010 data to be outdated? It would be useful to know what fraction of the surface altitude errors are due to ice sheet dynamics vs. altimetry errors.**

To quantify the contributions of ice sheet dynamics vs. altimetry errors a more detailed analysis would be needed. We assume regions with a more uniform difference (e.g. region 1) to be more influenced by ice sheet dynamics and regions with large difference with both signs (e.g. region 2, east coast of Greenland) to be more influenced by altimetry errors. However as we mentioned above, attribution to these error sources is out of the scope of this paper.

---

## Author Comment (AC2)

**Reply to reviewer 2**

Jonas Hachmeister

June 10, 2022

We would like to thank the reviewer for their helpful comments. Below the specific comments are listed in **bold text**. Our replies are in standard black text and the changed passages from the paper are in blue.

**Title: Have you considered including "TROPOMI" in the title? I'm suggesting it for an improved visibility through search engines etc.**

We added 'TROPOMI' to the title of the paper which now reads:

On the influence of underlying elevation data on Sentinel-5 Precursor TROPOMI satellite methane retrievals over Greenland

**Introduction: This section is completely missing the motivation for the need to address the elevation (or surface pressure) sensitivity of the retrieval and thus an improved elevation model. Since this is the content of the paper, I propose to introduce the topic in the introduction. Other high-latitude retrieval challenges have been mentioned (dark surfaces); perhaps also mention the elevation sensitivity there (I would also recommend mentioning the solar zenith angle limitations at high latitudes), and then add a paragraph, perhaps after the 3rd paragraph in introduction, about what you are addressing in this paper, along with relevant background on GMTED2010 (complementing the request by Reviewer 1 here). Applicable text has already been written in several other parts of the manuscript.**

We added the challenges of high solar zenith angles and the sensitivity to elevation data to the introduction:
Additionally the high solar zenith angles provide challenging measurement conditions. Furthermore the satellite retrievals depend on knowledge of the surface elevation e.g. for the calculation of surface pressure. The exact use of elevation data depends on the retrieval algorithm, however both datasets we investigate in this paper report a 1% error in the retrieved $XCH_4$ (about 20 ppb) for a 1% error in the surface pressure. This could lead to problems due to the use of inaccurate elevation data.

We also added a paragraph explaining what is addressed in the paper:

In this paper we investigate noticeable features in the maps of retrieved $XCH_4$ over Greenland which can be seen both in the operational S5P $XCH_4$ product and the S5P WFMD product. For this we investigate the digital elevation model (DEM) used in both retrievals, namely the Global Multi-resolution Terrain Elevation Data (GMTED2010) and compare it to new elevation data from the ICESat-2 satellite mission.

**Sect. 2.1.2 (and also 2.1.1 as applicable): I suggest to add information on the filtering (quality-screening) of the data, in particular because in e.g. Fig. 10 caption you refer to an updated quality filtering. You also most likely quality-screen the data before gridding so it is important to mention the qa_value criteria in 2.1.1 also.**

We added information on the quality screening to both sections.

Section 2.1.1:

The product includes a quality assurance value (qa) which is a continuous quality descriptor ranging from 0 (no data) to 1 (full quality data). As recommended in the product user guide [citation] we exclude data with qa< 0.5.

Section 2.1.2:

In another post-processing step the data is quality filtered using a machine learning approach based on a random forest classifier [citation]. We use data with a quality flag qf= 0 (good) and don't include data with qf= 1 (potentially bad).

**Sect. 2.1.2: This is more of a question than a comment or suggestion: could steep elevation changes (especially at high latitudes where the SZA are large) also have an effect on the retrievals through casting shadows? Likely this is much less significant; I was just looking at Fig. 2 where one can see different XCH4 anomalies in the northern coast of Greenland compared to elsewhere in the coast.**

Yes, this is indeed possible, however we assume these effects to be less significant. We note that the SZA is limited to 75° for the WFMD product and that the surface roughness is part of the product which can help with identifying the slopes, this would allow to filter the affected pixels. We have no definitive answer to these questions and plan to look into it in the future.

**Sect. 3.1: For the calculation of the 7-day methane anomaly, could you please specify how you do the gridding; is it only based on the centre coordinates of each pixel?**

Yes, the gridding is only based on the centre coordinate of each pixel. This information has been added to the manuscript.

We define the 7-day $XCH_4$ anomaly as follows: First we calculate the daily mean $XCH_4$ for every gridcell, where the gridding is based only on the centre coordinate of each pixel.

**Sect. 4.5 and Conclusions: I assume that the "preliminary version of the updated WFMD product" is indeed a preliminary reprocessing of the WFMD retrieval (i.e. considers also the updated reference spectra corresponding to the updated elevation information) and not limited to postprocessing corrections based on the linear relationship shown in the paper. Could you please specify this part in the paper?**

This is correct. The linear relationship found in the paper is not used in the postprocessing. We specified this in the updated manuscript:

Finally, we present a preliminary version of an updated WFMD product which is reprocessed using the Greenland DEM [citation] from instead of GMTED2010. Furthermore the quality filter is refined using additional ocean data in the training of the random forest classifier (see [citation]) (18 million added scenes compared to v1.5 equally distributed over 30 days) to reduce scenes with residual cloudiness in particular over the Arctic ocean in summer.

**Conclusions: Is the updated DEM recommended also for the retrievals of other atmospheric gases? Please specify.**

Depending on the retrieval strategy of the target gas in question, inaccurate DEM data will impact the retrieved column of other products as well. We recommend the usage of up-to-date and precise DEMs in all algorithms which rely on elevation data. While the magnitude of the errors may vary (or not be significant at all), we advise to use the most accurate data available to ensure highest possible quality of the resulting data products.